# Association of Serum Proteases and Acute Phase Factors Levels with Survival Outcomes in Patients with Colorectal Cancer

**DOI:** 10.3390/cancers16132471

**Published:** 2024-07-06

**Authors:** Tadeusz Sebzda, Jakub Karwacki, Anna Cichoń, Katarzyna Modrzejewska, Jerzy Heimrath, Mirosław Łątka, Jan Gnus, Jakub Gburek

**Affiliations:** 1Department of Pathophysiology, Wroclaw Medical University, 50-368 Wroclaw, Poland; tadeusz.sebzda@umw.edu.pl; 2University Center of Excellence in Urology, Department of Minimally Invasive and Robotic Urology, Wroclaw Medical University, 50-556 Wroclaw, Poland; 3Regional Specialist Hospital of St. Barbara, 41-200 Sosnowiec, Poland; anna.bajor@int.pl; 4University Hospital in Wroclaw, 50-556 Wroclaw, Poland; katarzyna.modrzejewska97@gmail.com; 5Collegium Witelona, 59-220 Legnica, Poland; jerzy.heimrath@collegiumwitelona.com; 6Department of Biomedical Engineering, Wroclaw University of Science and Technology, 50-370 Wroclaw, Poland; miroslaw.latka@pwr.edu.pl; 7Department of Physiotherapy, Wroclaw Medical University, 50-355 Wroclaw, Poland; jan.gnus@umw.edu.pl; 8Department of Pharmaceutical Biochemistry, Wroclaw Medical University, 50-556 Wroclaw, Poland

**Keywords:** colorectal cancer, cathepsin B, leukocytic elastase, sialic acid, C-reactive protein, acute phase factors, survival analysis

## Abstract

**Simple Summary:**

Colorectal cancer (CRC) presents a significant global health challenge, necessitating advancements in early detection and treatment strategies. This study aimed to identify the serum biomarkers associated with worse survival outcomes in CRC patients, focusing on cathepsin B (CB), leukocytic elastase (LE), total sialic acid (TSA), and others. A cohort of 185 CRC patients and 35 healthy controls underwent comprehensive testing, and statistical analyses were employed to identify significant correlations. The results revealed significant associations between CB (*p* = 0.04), LE (*p* = 0.01), TSA (*p* = 0.008), and survival outcomes. However, no significant associations were noted for other markers. Multivariate analysis found the correlation of LE, TSA, and ATA with survival (*p* = 0.041). Further research is needed to validate these findings and discover additional indicators.

**Abstract:**

Colorectal cancer (CRC) represents a substantial burden on global healthcare, contributing to significant morbidity and mortality worldwide. Despite advances in screening methodologies, its incidence remains high, necessitating continued efforts in early detection and treatment. Neoplastic invasion and metastasis are primary determinants of CRC lethality, emphasizing the urgency of understanding underlying mechanisms to develop effective therapeutic strategies. This study aimed to explore the potential of serum biomarkers in predicting survival outcomes in CRC patients, with a focus on cathepsin B (CB), leukocytic elastase (LE), total sialic acid (TSA), lipid-associated sialic acid (LASA), antitrypsin activity (ATA), C-reactive protein (CRP), and cystatin C (CC). We recruited 185 CRC patients and 35 healthy controls, assessing demographic variables, tumor characteristics, and 7 serum biomarker levels, including (1) CB, (2) LE, (3) TSA, (4) LASA, (5) ATA, (6) CRP, and (7) CC. Statistical analyses included ANOVA with Tukey’s post hoc tests and MANOVA for continuous variables. Student’s *t*-test was used for dependent samples, while non-parametric tests like Mann–Whitney U and Wilcoxon signed-rank tests were applied for variables deviating from the normal distribution. Categorical variables were assessed using chi-square and Kruskal-Wallis tests. Spearman’s rank correlation coefficient was utilized to examine variable correlations. Survival analysis employed the Kaplan–Meier method with a log-rank test for comparing survival times between groups. Significant associations were observed between CB (*p* = 0.04), LE (*p* = 0.01), and TSA (*p* = 0.008) levels and survival outcomes in CRC patients. Dukes’ classification stages also showed a significant correlation with survival (*p* = 0.001). However, no significant associations were found for LASA, ATA, CRP, and CC. Multivariate analysis of LE, TSA, and ATA demonstrated a notable correlation with survival (*p* = 0.041), notwithstanding ATA’s lack of significance in univariate analysis (*p* = 0.13). CB, LE, and TSA emerged as promising diagnostic markers with prognostic value in CRC, potentially aiding in early diagnosis and treatment planning. Further research is needed to validate these findings and explore additional prognostic indicators.

## 1. Introduction

Colorectal cancer (CRC) accounted for nearly 2 million new cases and approximately 1 million deaths in 2020, representing 10.7% of all new cancer cases and 9.5% of all cancer-related deaths worldwide [1]. Despite the increasing role of screening and the growing number of cases detected through the screening strategy rather than symptom presentation [2,3], CRC persists as a significant global healthcare concern, emphasizing the critical importance of early diagnosis [4].

The neoplastic invasion and metastasis represent the pivotal aspects that render the disease lethal [5]. As cancer-transformed cells migrate, they encounter morphological barriers such as the basement membrane and connective tissue. Specific proteases and acute phase factors assume a crucial role in surmounting these barriers during this process [6,7,8,9,10,11]. Under physiological conditions, cathepsins are predominantly intracellular enzymes involved in protein turnover and the degradation of exogenous proteins absorbed through endocytosis [12,13,14,15]. However, their release from cancer cells and expression on the cell membrane facilitate tumor cell invasion and metastasis [16]. Clinical observations show a significant increase in serum cathepsin B (CB) activity in patients with various malignant tumors, including CRC, indicating its potential role as a sensitive marker for disease progression [9,17]. Leukocytic elastase (LE), primarily found in neutrophil azurophilic granules, aids in phagocytosis alongside other enzymes and reactive oxygen species [18,19,20]. It contributes to tissue remodeling, cytokine modulation, and extracellular matrix degradation. Physiological regulation involves alpha-1 antitrypsin inhibitor (AAT), preventing excessive proteolysis [19]. Elevated LE levels, often as LE-AAT complexes, along with reduced AAT levels or imbalance, signal inflammation and neutrophil activation and can lead to cancer progression [20,21,22,23]. Sialic acid (N-acetylneuraminic acid, NANA) is a sugar component that is crucial for cell physiology and immune response [24,25]. Elevated levels are linked to metabolic disorders and various cancers, indicating its potential as a cancer biomarker [26]. Glycoproteins that are abundant in sialic acids are commonly observed in metastatic cancer. This elevation in sialoglycoproteins and sialoglycolipids within tumors primarily stems from the augmented breakdown of cancer cells, alongside increased synthesis and release of glycoconjugates containing sialic acid [27,28,29,30,31]. C-reactive protein (CRP) and cystatin C (CC) are also implicated in cancer progression and metastasis, highlighting the importance of investigating their role in CRC as well [32,33,34].

The study of patient survival outcomes in CRC is crucial as it provides insights into the effectiveness of current treatment strategies and helps identify areas needing improvement. Understanding survival outcomes enables physicians to better predict disease progression, tailor treatments to individual patients, and ultimately improve prognosis and quality of life. Biomarkers play a pivotal role in this context, offering potential tools for early detection, prognosis, and monitoring therapeutic responses [35]. Recent studies suggest that a combination of gene mutations (such as KRAS, BRAF, and p 53) and epigenetic changes (like DNA methylation) significantly contribute to CRC development, guiding targeted therapies and improving patient outcomes through molecular testing [36]. Though some biomarkers are well known and tested in clinical settings, others appear promising but have not yet been investigated in such contexts.

This study represents an extension of our prior investigation into the same subject matter [7]. However, the current study places particular emphasis on analyzing patient survival outcomes and determining potential associations with the investigated parameters. The objective of this study was to investigate potential correlations between the serum levels or activity of specific biochemical parameters, including CB, LE, TSA, lipid-associated sialic acid (LASA), antitrypsin activity (ATA), CRP, and CC, and survival outcomes in patients diagnosed with colorectal adenocarcinoma.

## 2. Materials and Methods

### 2.1. Patient Population

The study population comprises 185 patients with colorectal adenocarcinoma enrolled at the Lower Silesian Oncology Center and the Provincial Specialist Hospital in Wroclaw. Patients were comprehensively assessed for demographic variables, including age and sex, as well as the anatomical site of the neoplastic lesion. Furthermore, histopathological grading of the neoplastic cells (G) and clinical staging according to the Dukes’ classification were performed [37]. The Dukes’ classification system delineated stages as follows: Dukes A denoted invasion into, but not through, the bowel wall; Dukes B indicated invasion through the bowel wall, penetrating the muscle layer but without lymph node involvement; Dukes C signified lymph node involvement; and Dukes D represented widespread metastases. Blood serum samples were obtained preoperatively from each patient. All CRC patients underwent surgery. None of the patients received adjuvant therapy as part of their management. However, it was challenging to track follow-up data regarding salvage or palliative therapy. Consequently, we included patients who received such therapies in the study. The control group comprised 35 patients from the Lower Silesian Oncology Center and the Provincial Specialist Hospital in Wroclaw, who were admitted for reasons other than CRC, such as functional gastroenterological disorders.

### 2.2. Biochemical Measurements

The study examined 7 serum markers, which comprised (1) CB, (2) LE, (3) TSA, (4) LASA, (5) ATA, (6) CRP, and (7) CC. The blood serum underwent several tests using different methodologies. CB levels were determined using fluorogenic substrates following the Barrett method [38]. LE, in conjunction with AAT, was measured via immunoenzymatic analysis employing the MERCK test. TSA levels were assessed colorimetrically using the periodate-resorcinol method introduced by Jourdian et al. [39]. LASA was quantified colorimetrically, following the procedure outlined by Tautu et al. [40]. Serum ATA (or antitrypsin capacity) in blood plasma was evaluated colorimetrically against trypsin utilizing the methodology proposed by Warwas et al. and Dietz et al. [41,42]. CRP levels were determined immunoturbidimetrically using the MERCK test. CC concentrations were assessed via an immunoturbidimetric assay utilizing the DAKO Cystatin C PET Kit (Denmark).

### 2.3. Statistical Analysis

Continuous variables were characterized using the mean, standard deviation (SD), and sample size (*n*). Statistical analysis of the data employed for continuous variables includes one-way analysis of variance (ANOVA) with Tukey’s post hoc tests and multivariate analysis of variance (MANOVA). For dependent samples, the Student’s *t*-test was utilized. Non-parametric tests, such as the Mann–Whitney U test for independent samples and the Wilcoxon signed-rank test for dependent samples, were applied to variables deviating from the normal distribution. Categorical or dichotomous variables were assessed using the chi-square and Kruskal–Wallis tests. Additionally, the correlation between variables was examined using Spearman’s rank correlation coefficient. The determination of marker threshold values was conducted utilizing the receiver operating characteristic (ROC) curve method, with comprehensive details provided in our initial publication on this topic [7]. The ROC analysis for the multiparameter model comprising LE, TSA, and ATA is presented in the Appendix A. Survival analysis of patient groups employed the Kaplan–Meier method, with the log-rank test used for comparing survival times between two groups. A significance level of *p* < 0.05 was adopted for statistical evaluation, with non-significance denoted as NS. Results differing at significance levels of *p* < 0.01 and *p* < 0.001 were also presented. Statistical analysis was performed using the statistical software package Statistica version 10.

## 3. Results

The demographic and clinical data of the 185 CRC patients are outlined in Table 1. The control cohort comprised 35 patients, with a median age of 61 years (range: 19–85), consisting of 19 men (54.3%) and 16 women (45.7%). Among the CRC patients, 98 (53.0%) were male and 87 (47.0%) were female, with a median age of 63 years (range: 18–86). The assessment of tumor localization revealed that 77 patients (41.6%) had lesions in the colon, 37 (20.0%) in the sigmoid colon, and 71 (38.4%) in the rectum. Histological grading demonstrated 8 patients (4.3%) with G1 disease, 103 (55.7%) with G2, and 74 (40.0%) with G3 tumor. According to the Dukes’ classification, 22 patients (11.9%) were categorized as group A, 52 (28.1%) as group B, 72 (38.9%) as group C, and 39 (21.1%) as group D.

In the analysis of 5-year patient survival rates, considering the degree of differentiation of tumor cells (G2 and G3), the significance level was found to be *p* = 0.042 (Appendix A). The relationship between patient survival and the stage of tumor advancement according to the Dukes classification reached a high level of significance with *p* = 0.001 (Appendix A). The survival curve based on tumor localization (rectum, colon, and sigmoid) did not reveal statistically significant differences (Appendix A).

Table 2 displays the findings of the examined biochemical parameters, including their mean, SD, and sample size, encompassing both the colorectal cancer patient group and the control group. Our previous study [7] yielded analogous results. In the CRC group, differences in the values of the examined biochemical parameters in serum were observed compared to the control group. Despite notable changes in the levels and activities of the investigated factors, not all of these differences were statistically significant across all groups. Significance was observed between the CRC patient groups and the control group regarding the levels of CB, LE, TSA, and ATA.

Survival analysis using the Kaplan–Meier test was conducted for selected parameters, as depicted in Figure 1, Figure 2, Figure 3, Figure 4, Figure 5, Figure 6 and Figure 7. The analysis demonstrated the significance of CB levels (*p* = 0.04, Figure 1) in assessing survival in CRC patients, with those below the cutoff point showing significantly poorer survival times. Conversely, varying LE activity across Duke’s classification stages of CRC precludes its early diagnostic utility. However, LE confirms its prognostic value, with a cutoff point of 543 μg/L distinguishing survival outcomes significantly (*p* = 0.01, Figure 2) in CRC patients. Similarly, for TSA levels above 75.34 mg%, Kaplan–Meier analysis revealed significantly poorer survival outcomes (*p* = 0.008, Figure 3). Kaplan–Meier analysis for LASA, ATA, CRP, and CC showed no significant association with survival outcomes in the CRC group (Figure 4, Figure 5, Figure 6 and Figure 7). 

Additionally, survival analyses were conducted for all the aforementioned parameters to analyze the effect of various parameter combinations on the survival outcomes. One Kaplan–Meier test for the combination of LE, TSA, and ATA revealed a significant impact of these factors on survival in CRC patients (*p* = 0.041, Figure 8). The threshold for this multifactorial analysis has been set at −0.113, as determined by the ROC curve (Appendix A). Youden’s index was 0.55, indicating that this analysis has moderate accuracy. This means it is useful as it correctly identifies both positive and negative cases more often than it misses them.

The graphical summary of the association between the investigated biochemical parameters and survival outcomes is depicted in Figure 9. The same associations are presented as the tabular summary in Table 3.

## 4. Discussion

Several studies consistently demonstrate that elevated levels of CB are linked to unfavorable survival outcomes in CRC patients. Troy et al. [43] noted that elevated CB and cathepsin L (CL) activity ratios were associated with decreased survival in individuals with potentially treatable conditions. In early-stage disease, both CB and CL tumor/normal activity ratios exceeded 1, with gradual reductions observed as tumor stage advanced (*p* = 0.02 for CB). Moreover, patient survival in potentially curable cases exhibited an inverse relationship with both CB (*p* = 0.007) and CL (*p* = 0.001) activity ratios. Kos et al. [44] demonstrated in their survival analysis that colorectal cancer (CRC) patients with elevated serum CB levels showed notably diminished survival probability at the cutoff value of 9.4 ng/ml. Furthermore, individuals with heightened serum levels of CB and carcinoembryonic antigen (CEA) had markedly reduced survival rates (relative hazard ratio [HR] of 2.2; 95% confidence interval, 1.5–3.2; *p* < 0.0001) compared to those with lower levels of both molecules. In a similar vein, Campo et al. [45] noted a correlation between elevated CB expression and advanced disease stage, leading to shortened patient survival. Increased enzyme expression in tumor stromal cells corresponded with neoplastic advancement. Moreover, high levels of CB expression in tumor epithelial cells were associated with significantly reduced patient survival. Additionally, Chan et al. [46] reinforced these findings by demonstrating that the presence of CB in tumors was linked to a heightened risk of both disease-specific and overall mortality. However, CB expression did not correlate with disease stage (*p* = 0.19). Participants with CB-positive tumors had a multivariate HR of 1.99 (95% confidence interval [CI] 1.19–3.34) for disease-specific mortality and 1.71 (95% CI 1.16–2.50) for overall mortality compared to those with CB-negative tumors. Collectively, these studies underscore the significant role of cathepsin B levels/activity in predicting survival outcomes among colorectal cancer patients.

Research indicates that LE levels and activity may impact the prognosis of colorectal cancer (CRC) patients. In our investigation, CRC patients with lower LE levels exhibited poorer survival outcomes compared to those with higher levels, with a threshold of 534 μg/L (Figure 2). Ho et al. [21] observed elevated neutrophil elastase (NE) expression in CRC patients, suggesting its potential as a diagnostic marker and therapeutic target. However, Berry et al. [47] reported that high levels of tumor-associated neutrophils, which release NE, were associated with improved overall survival in stage II CRC patients. Conversely, Chiang et al. [48] and Zhang et al. [49] found that an elevated neutrophil-to-lymphocyte ratio, influenced by NE, correlated with worse outcomes in CRC patients. These findings underscore a nuanced relationship between LE and survival outcomes in CRC patients, underscoring the need for further investigation.

Both TSA and LASA are useful markers in the diagnosis and monitoring of various conditions, including CRC and other malignancies [6,24,50,51,52,53,54]. In our study, the Kaplan–Meier analyses of patients with CRC revealed statistically significant differences in TSA levels (*p* = 0.008), but not in LASA levels (*p* = 0.89). TSA and LASA has been found to be significantly elevated in the serum of CRC patients, with higher levels correlating with more advanced stages of the disease [40,55]. Preoperative serum TSA levels have been identified as a potential prognostic factor for tumor recurrence in CRC, with the TSA/total protein (TP) ratio showing promise as a marker for high-risk patients [56].

The role of ATA in the CRC is complex and multifaceted. Studies have shown that the expression of pancreatic secretory trypsin inhibitor (PSTI) is widespread in CRC, particularly in advanced cases [57]. This suggests a potential role in tumor development. The presence of antitrypsins in CRC cells and their metastatic foci further supports their potential role in tumor progression [58]. However, the exact mechanisms and implications of these findings in the context of colorectal cancer require further investigation.

Remarkably, the multivariate analysis of LE, TSA, and ATA demonstrated a notable correlation with survival among CRC patients (*p* = 0.041), notwithstanding ATA’s lack of significance in the univariate analysis (*p* = 0.13). This indicates the potential for ATA to be linked with survival outcomes in larger cohorts in future investigations. Consequently, LE and TSA have solidified their significance as the most influential biochemical parameters in our study.

CRP, a marker of systemic inflammation, has consistently been linked to poor survival outcomes in colorectal cancer. Toiyama et al. [59] identified CRP positivity as an independent prognostic marker in stage I-III CRC, especially in cases with inadequate lymph node retrieval. Van de Poll et al. [60] further supported this, demonstrating that elevated CRP concentrations were associated with decreased survival in patients with colorectal peritoneal carcinomatosis. Similarly, Wong et al. [61] found that elevated preoperative CRP levels predicted poor outcomes in patients undergoing curative resection for colorectal liver metastases. These findings underscore the potential value of CRP as a prognostic tool for colorectal cancer. CC, a cysteine protease inhibitor, has been identified as a potential tumor marker for CRC. Its role in CRC is further supported by the finding that cystatin SN, a member of the cystatin family, is highly expressed in CRC cells [62]. This suggests that cystatin C may play a role in the progression and metastasis of CRC. However, the specific mechanisms by which cystatin C contributes to CRC development and progression remain to be fully elucidated. Our study did not reveal correlations between CRP and CC levels and survival outcomes in CRC patients.

In terms of limitations, it should be noted that not all patients had complete data available. This was performed deliberately to include the maximum amount of data possible in order to achieve our study objectives and provide comprehensive and thorough results. Another limitation is the relatively small sample size, which may have contributed to the lack of statistical significance in some associations, although several approached significance. Larger study cohorts would likely uncover more statistically significant associations. Additionally, a significant limitation of this study is the necessity to use Kaplan–Meier curves instead of multivariable analysis, such as Cox regression. This choice was driven by our inability to retrieve all clinical data due to the restructuring of the hospital where the patients were admitted. The loss of some clinical data prevented us from performing a more comprehensive multivariable analysis. Although Kaplan–Meier curves provide valuable insights into survival outcomes and allow us to visualize the association between specific biochemical parameters and survival, they do not account for the potential confounding effects of multiple variables. Despite these constraints, we believe that our findings contribute significantly to the understanding of survival outcomes in CRC patients and highlight the importance of further research using more complete datasets.

## 5. Conclusions

The Kaplan–Meier analyses of patients with CRC revealed statistically significant differences in CB (*p* = 0.04), LE (*p* = 0.01), and TSA (*p* = 0.008) levels. Survival analysis considering tumor cell differentiation (G2 vs. G3) yielded a significance level of *p* = 0.042. Additionally, significant associations were found between patient survival and Dukes’ classification stages (*p* = 0.001), while survival curves based on tumor location showed no statistically significant differences. Our study identified CB, LE, and TSA as promising diagnostic indicators in CRC, with prognostic capabilities to forecast survival outcomes in affected patients.

## Figures and Tables

**Figure 1 cancers-16-02471-f001:**
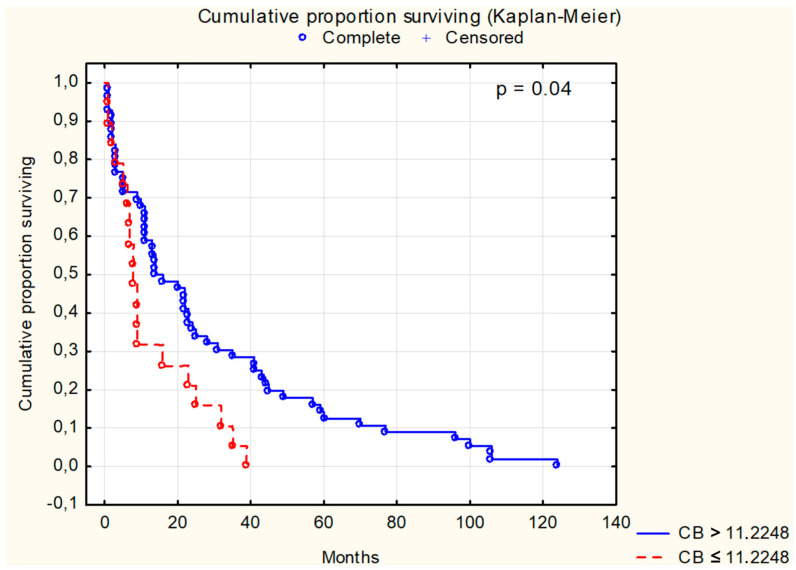
The survival analysis of patients with colon adenocarcinoma at serum cathepsin B threshold of 11.2248 mU/L. The Kaplan–Meier test was used to perform the survival analysis. *p*: statistical significance; CB: cathepsin B.

**Figure 2 cancers-16-02471-f002:**
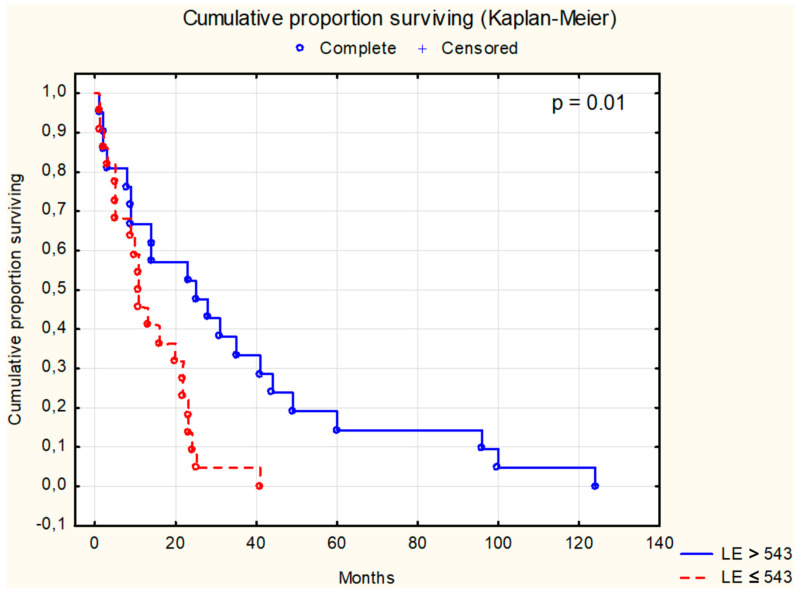
The survival analysis of patients with colon adenocarcinoma at serum leukocytic elastase threshold of 534 μg/L. The Kaplan–Meier test was used to perform the survival analysis. *p*: statistical significance; LE: leukocytic elastase.

**Figure 3 cancers-16-02471-f003:**
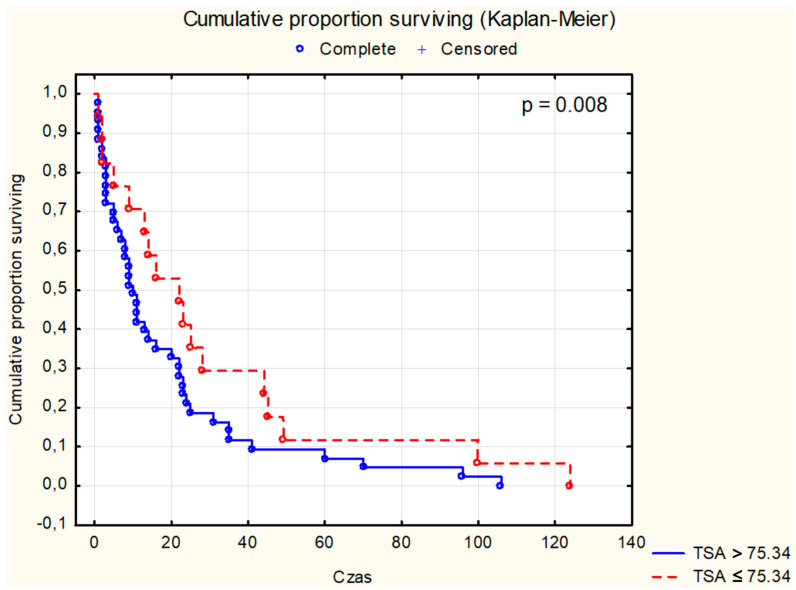
The survival analysis of patients with colon adenocarcinoma at serum total sialic acid threshold of 75.34 mg%. The Kaplan–Meier test was used to perform the survival analysis. *p*: statistical significance; TSA: total sialic acid.

**Figure 4 cancers-16-02471-f004:**
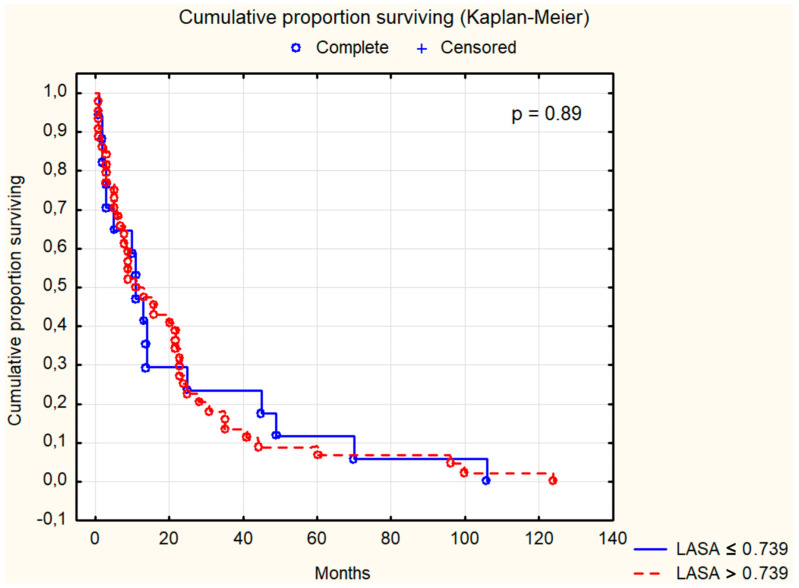
The survival analysis of patients with colon adenocarcinoma at serum lipid-associated sialic acid threshold of 0.739 mg%. The Kaplan–Meier test was used to perform the survival analysis. *p*: statistical significance; LASA: lipid-associated sialic acid.

**Figure 5 cancers-16-02471-f005:**
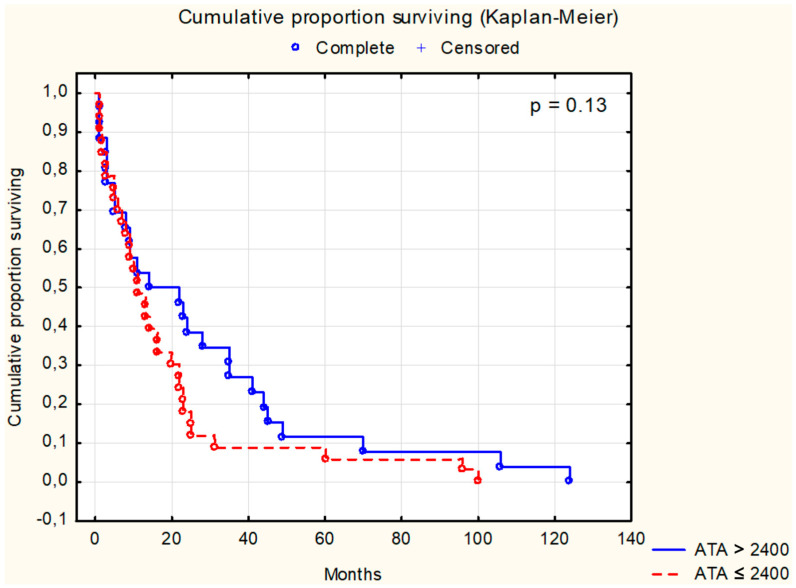
The survival analysis of patients with colon adenocarcinoma at serum antitrypsin activity threshold of 2400 U/mL. The Kaplan–Meier test was used to perform the survival analysis. *p*: statistical significance; ATA: antitrypsin activity.

**Figure 6 cancers-16-02471-f006:**
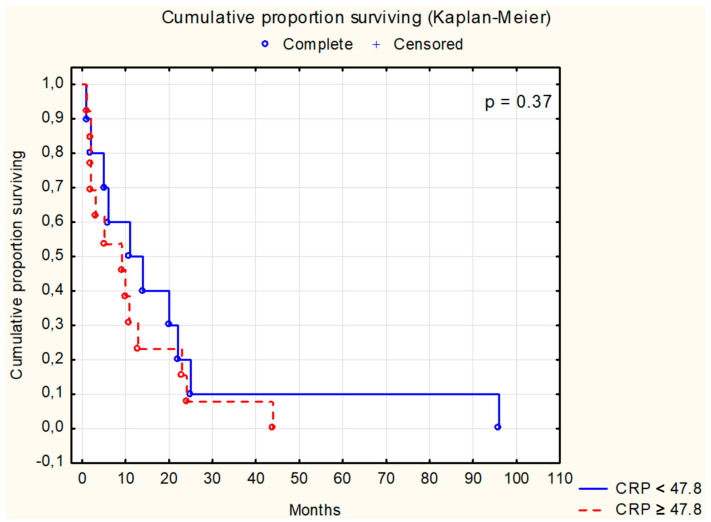
The survival analysis of patients with colon adenocarcinoma at serum C-reactive protein threshold of 47.8 mg/L. The Kaplan–Meier test was used to perform the survival analysis. *p*: statistical significance; CRP: C-reactive protein.

**Figure 7 cancers-16-02471-f007:**
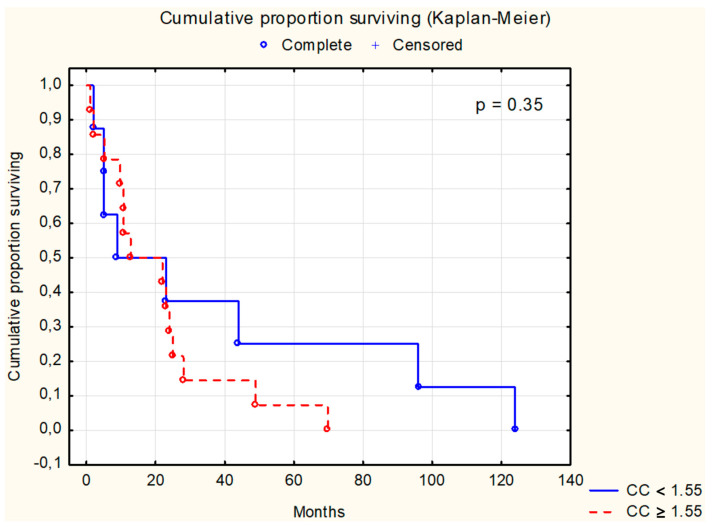
The survival analysis of patients with colon adenocarcinoma at serum cystatin C threshold of 1.55 mg/L. The Kaplan–Meier test was used to perform the survival analysis. *p*: statistical significance; CC: cystatin C.

**Figure 8 cancers-16-02471-f008:**
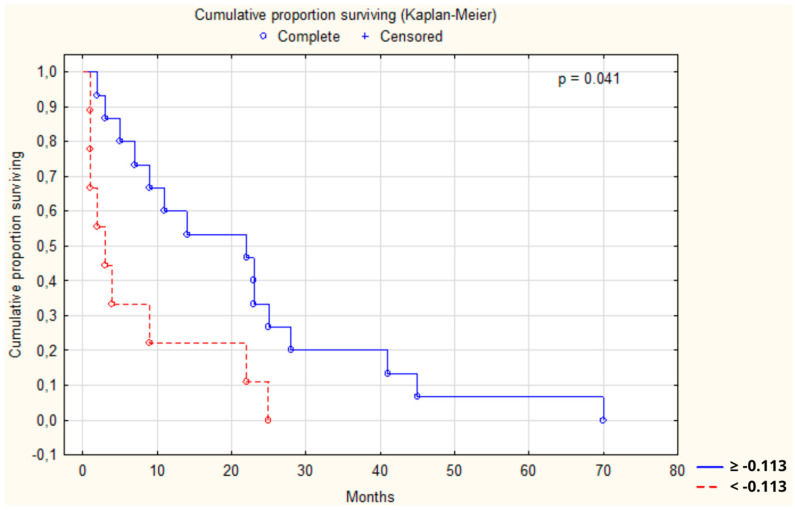
The survival analysis of patients with colon adenocarcinoma at combined threshold of −0.113 mg/L for mulitparameter model of leukocytic elastase, total sialic acid, and antitrypsin activity. The Kaplan–Meier test was used to perform the survival analysis. *p*: statistical significance.

**Figure 9 cancers-16-02471-f009:**
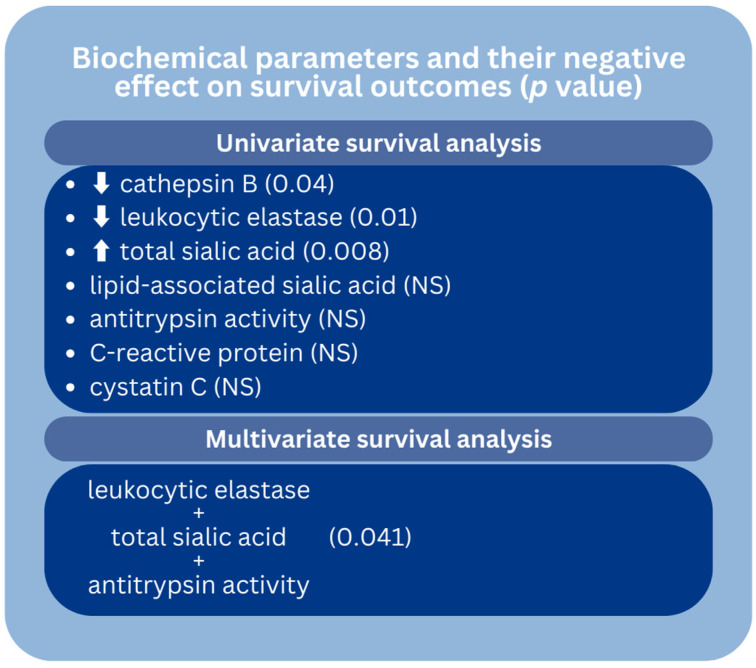
The graphical summary of the association of investigated biochemical factors and survival outcomes in colorectal adenocarcinoma patients. Our study revealed that decreased levels of cathepsin B and leukocytic elastase, along with elevated levels of total sialic acid, were associated with poorer survival outcomes. However, the association of lipid-associated sialic acid, antitrypsin activity, C-reactive protein, and cystatin C with survival outcomes was found to be non-significant. In multivariate survival analysis, only a combination of leukocytic elastase, total sialic acid, and antitrypsin activity showed a correlation with survival outcomes. *p*: probability; NS: non-significant.

**Table 1 cancers-16-02471-t001:** Characteristics of 185 patients with colorectal adenocarcinoma.

**Sex,** ***n*** **(%)**	
Men	98 (53.0)
Women	87 (47.0)
**Age,** **median (range)**	63 (18–86)
**Anatomical** **location, *n* (%)**	
Colon	77 (41.6)
Sigmoid colon	37 (20.0)
Rectum	71 (38.4)
**Histological** **grade, *n* (%)**	
G1	8 (4.3)
G2	103 (55.7)
G3	74 (40.0)
**Dukes’** **classification, *n* (%)**	
Dukes A	22 (11.9)
Dukes B	52 (28.1)
Dukes C	72 (38.9)
Dukes D	39 (21.1)

*n*: number of patients; G: grade.

**Table 2 cancers-16-02471-t002:** Characteristics of serum parameters in CRC patients and in the control group.

Biochemical Arameter, Unit	*n*	CRC Patients (Mean ± SD)	*n*	Control Group Patients (Mean ± SD)	*p* Value
CB, mU/L	185	16.1 ± 8.8	35	11.4 ± 6.5	<0.050
LE, μg/L	51	875.1 ± 597.9	30	379.1 ± 187.3	<0.001
TSA, mg%	71	98.9 ± 30.8	31	71.4 ± 15.1	<0.001
LASA, mg%	68	0.68 ± 0.33	35	0.69 ± 0.28	NS
ATA, U/mL	74	3211.4 ± 1504.1	29	2015.9 ± 689.6	<0.001
CRP, mg/L	34	59.3 ± 43.5	23	12.9 ± 10.90	NS
CC, mg/L	34	2.17 ± 2.48	15	1.10 ± 0.24	NS

*n*: number of patients; CRC: colorectal cancer; SD: standard deviation; *p*: probability; CB: cathepsin B; LE: leukocytic elastase; TSA: total sialic acid; LASA: lipid-associated sialic acid; NS: nonsignificant; ATA: antitrypsin activity; CRP: C-reactive protein; CC: cystatin C.

**Table 3 cancers-16-02471-t003:** The tabular summary of the association between investigated biochemical factors and survival outcomes in colorectal adenocarcinoma patients.

**Univariate Survival Analysis**	
**Biochemical Parameters**	***p* Value**
Cathepsin B	0.04
Leukocytic elastase	0.01
Total sialic acid	0.008
Lipid-associated sialic acid	NS
Antitrypsin activity	NS
C-reactive protein	NS
Cystatin C	NS
**Multivariate Survival Analysis**	
**Biochemical Parameters**	***p*** **Value**
Leukocytic elastase, total sialic acid, and antitrypsin activity	0.041

*p*: probability; NS: non-significant.

## Data Availability

The data supporting this study’s findings are accessible upon request. Please contact the corresponding author to obtain the dataset. However, please note that not all data is available due to the restructuring of the hospital where the patients were admitted.

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
