# Peer review of "Association of Serum Proteases and Acute Phase Factors Levels with Survival Outcomes in Patients with Colorectal Cancer"

_cancers, 2024, doi:10.3390/cancers16132471_

Round 1

Reviewer 1 Report

Comments and Suggestions for Authors

This is a well written article. One important question is about the cut-off points for the biomarkers that underpin the survival curves. Why were those specific cut-offs chosen? For example why was Cathepsin B's cut-off 11.2248? This is not clearly explained in the article but it is really quite important as it is the key to the survival analyses. If you adjust the cut-off points for the biomarkers even slightly there could be a very different result. 

Author Response

The reply is in the attached PDF file.

Reviewer 2 Report

Comments and Suggestions for Authors

Association of Serum Proteases and Acute Phase Factors Levels with Survival Outcomes in Patients with Colorectal Cancer, examines several markers for use in prognosis of CRC.  As is often the case, some of the proposed markers were not of use and 3 were found to be of suggestive value, individually, possibly as the authors suggest, due to the small number of subjects .  When examined as a set the prognostic value was just significant.  I would like to see a ROC analysis for the LE, TSA, and ATA set.  

Author Response

The reply is in the attached PDF file.

Reviewer 3 Report

Comments and Suggestions for Authors

This study by Tadeusz Sebzda et al. explored the potential of serum biomarkers in predicting survival outcomes in colorecatal cancer (CRC) patients, with a focus on cathepsin B (CB), leukocytic elastase (LE), total sialic acid (TSA), lipid-associated sialic acid (LASA), antitrypsin activity (ATA), C-reactive protein (CRP), and cystatin C 3(CC). The authors recruited 185 CRC patients and 35 healthy controls, assessing demographic variables, tumor characteristics, and serum biomarker levels. The authors observed significant associations were observed between CB, LE, and TSA levels and survival outcomes in CRC patients. The authors conclude that CB, LE, and TSA emerged as promising diagnostic markers with prognostic value in CRC, potentially aiding in early diagnosis and treatment planning.

The study question is valid, and the work is scientifically sound. The study methods are appropriate and adequately described. The manuscript is well written overall and adequate references are included. In my opinion, the authors of the manuscript have done a good job formulating the study plan and presenting their findings in clear manner. The manuscript can be improved by addressing the following concerns.

-       Introduction section may benefit from discussing in detail about the importance of studying patient survival outcomes in CRC and the role of biomarkers in this field.

-       Authors may want to comment on if the study was approved by institutional review board. 

-       How the control group was recruited.

-       Did authors study the presence of other variables that could have influenced the biomarker levels?

-       Use a table to present univariate and multivariate analyses findings.

Author Response

The reply is in the attached PDF file.

Reviewer 4 Report

Comments and Suggestions for Authors

This is a retrospective analysis of biomarkers in CRC.

Major

1. Were the cohort and the control of this study the same as your previous study (ref 7)? The results shown in the first part of this study were quite similar to your previous paper. Please clarify what is new in this study (survival analysis?) to avoid the criticism of double publications.

2. As for survival analysis, please perform the multivariable analysis including detailed clinical parameters and show hazard ratios.

3. Clinical data need to be desribed more in detail. Did all patients undergo surgery? Any adjuvant or palliative chemotherapy?

Minor

1. Please add p-values not just NS or p<0.05.

2. Please add number at risk in K-M analyses.

3. Please describe how the controls were selected.

Author Response

The reply is in the attached PDF file.

Round 2

Reviewer 2 Report

Comments and Suggestions for Authors

Add a comment about the ROC analysis to Results along with assessment of strength.

Author Response

Thank you very much for pointing this out. We have added a comment and the explanation to the Results section (lines 200-204: "The threshold for this multifactorial analysis...").

Reviewer 4 Report

Comments and Suggestions for Authors

The authors failed to provide precise data, which the reviewer believed mandatory for presentation of study results.

This is contradictory to your following statement.

Data Availability Statement: The data supporting this study's findings are accessible upon request. Please contact the corresponding author to obtain the dataset.

Author Response

Thank you very much for pointing it out.

1) We have added a comment on the accuracy of our multifactorial analysis (lines 200-204: The threshold...).

2) We have added a clearer version of data availability statement (The data supporting this study's findings are accessible upon request. Please contact the corresponding author to obtain the dataset. However, please note that not all data is available due to the restructuring of the hospital where the patients were admitted.)

Thank you very much for your valuable feedback and understanding. We appreciate your constructive comments, which have been vital in improving our manuscript.